# Breast Cancer Subtypes Underlying EMT-Mediated Catabolic Metabolism

**DOI:** 10.3390/cells9092064

**Published:** 2020-09-09

**Authors:** Eunae Sandra Cho, Nam Hee Kim, Jun Seop Yun, Sue Bean Cho, Hyun Sil Kim, Jong In Yook

**Affiliations:** 1Department of Oral Pathology, Oral Cancer Research Institute, Yonsei University College of Dentistry, Seoul 03722, Korea; SANDRA@yuhs.ac (E.S.C.); MIGO77@yuhs.ac (N.H.K.); YJS8714@yuhs.ac (J.S.Y.); chosuebean@gmail.com (S.B.C.); 2BK21 PLUS Project, Yonsei University College of Dentistry, Seoul 03722, Korea

**Keywords:** Snail, EMT, catabolic metabolism, breast cancer subtypes

## Abstract

Efficient catabolic metabolism of adenosine triphosphate (ATP) and reduced nicotinamide adenine dinucleotide phosphate (NADPH) is essentially required for cancer cell survival, especially in metastatic cancer progression. Epithelial–mesenchymal transition (EMT) plays an important role in metabolic rewiring of cancer cells as well as in phenotypic conversion and therapeutic resistance. Snail (SNAI1), a well-known inducer of cancer EMT, is critical in providing ATP and NADPH via suppression of several gatekeeper genes involving catabolic metabolism, such as phosphofructokinase 1 (PFK1), fructose-1,6-bisphosphatase 1 (FBP1), and acetyl-CoA carboxylase 2 (ACC2). Paradoxically, PFK1 and FBP1 are counter-opposing and rate-limiting reaction enzymes of glycolysis and gluconeogenesis, respectively. In this study, we report a distinct metabolic circuit of catabolic metabolism in breast cancer subtypes. Interestingly, PFKP and FBP1 are inversely correlated in clinical samples, indicating different metabolic subsets of breast cancer. The luminal types of breast cancer consist of the pentose phosphate pathway (PPP) subset by suppression of PFKP while the basal-like subtype (also known as triple negative breast cancer, TNBC) mainly utilizes glycolysis and mitochondrial fatty acid oxidation (FAO) by loss of FBP1 and ACC2. Notably, PPP remains active via upregulation of TIGAR in the FBP1-loss basal-like subset, indicating the importance of PPP in catabolic cancer metabolism. These results indicate different catabolic metabolic circuits and thus therapeutic strategies in breast cancer subsets.

## 1. Introduction

Malignant cancer cells continuously encounter altered environmental conditions of nutrient deficiency, hypoxia, and acidosis that challenge its survival. Trade-offs between proliferation and survival are essential for cancer cells to adapt and settle in their dynamic environment [1]. Cancer cells pursue this balance via metabolic reprogramming of biosynthesis, bioenergetics, and redox homeostasis [1,2,3]. The proliferative and survival phenotypes, also termed fast and slow life history, respectively, require different balances of these metabolic products. Proliferation requires extensive amounts of biosynthesis, thus avoiding complete consumption of carbons during bioenergy acquisition, whereas survival is promoted through adequate bioenergy supply enriched by the mitochondria. Redox power is essential for both fatty acid synthesis (FAS) and oxidative stress, and its homeostasis is gained by allosteric regulation of its metabolite based on situational requirements [4,5,6].

Epithelial–mesenchymal transition (EMT) is a cellular process presented in embryogenesis, wound healing, and cancer whereby mesenchymal traits of loss of adherence and polarity from epithelial cells. In addition to phenotypic transition, EMT is involved in stemness, invasiveness, migration ability, apoptotic resistance, and metabolic reprogramming in cancers [7,8,9,10,11]. The mesenchymal phenotype has advantages for dissemination, yet structural plasticity is not the only conversion required for entrance and migration within circulation. Metastasis is highly inefficient, primarily because cancer cells lose glucose transport caused due to cellular detachment to adjacent epithelial cells or extracellular matrix [12,13,14,15]. To overcome metastatic inefficiency, cancer cells must conduct oncogenic metabolic reprogramming to compensate for adenosine triphosphate (APT) generation and reduce its consequence, oxidative stress.

The ATP and reduced nicotinamide adenine dinucleotide phosphate (NADPH) are essential catabolic metabolites under starved environment, and are controlled by interconnected glycolysis, gluconeogenesis, pentose phosphate pathway (PPP), and mitochondrial activity such as fatty acid oxidation (FAO) and tricarboxylic acid (TCA) cycles (Figure 1). Recently, several metabolic enzymes have been identified as target genes of Snail transcriptional repressor including phosphofructokinase, platelet (PFKP), fructose-1,6-bisphosphatase 1 (FBP1), and acetyl-CoA carboxylase (ACC2) [16,17,18]. Snail repressor directly binds to the E-boxes located in proximal promoter of those target genes and Snail abundance is inversely correlated [16,17,18]. Interestingly, these Snail target genes are critical enzymes involving NADPH and ATP generation under glucose-starved environment, supporting the important role of Snail in catabolic metabolism during metastatic cancer progression. In this article, we investigate the coordination of metabolic pathways expressed in breast cancer molecular subtypes underlying EMT-regulated catabolic metabolism.

## 2. Methods

Publicly available mRNASeq data of human breast cancer samples (BRCA) was downloaded (https://gdac.broadinstitute.org). The illuminahiseq_rnaseqv2-RSEM_genes_normalized (MD5) was log2 transformed and the relative transcript abundance of PFKP, ACC2 (ACACB), FBP1, and TIGAR were compared using adjusted *p*-value (Benjamini–Hochberg). For an unsupervised hierarchical cluster analysis, Ward linkage method was used together with the Pearson distance for both sample and gene clustering. Among 1093 BRCA patient samples with subtype information, mutational status of BRCA1/2, PIK3CA, and p53 were available in 973 samples for hierarchical cluster analysis. The statistical significance of PFKP, ACC2, FBP1, and TIGAR according to the breast cancer subtypes was determined by Tukey’s HSD (honestly significant difference) test.

## 3. Glycolysis and Gluconeogenesis

Biosynthesis and related anabolic pathways have been investigated in cancer primarily in the context of targeting and controlling the vast growth potential of the tumor. Glucose serves as a primary energy source for proliferation in both normal and cancer cells in ideal glucose-rich conditions. It is not surprising that aerobic glycolysis is evident in proliferating cancer cells, as described early by Warburg [19]. However, the glycolytic pathway is highly inefficient in terms of ATP generation compared to oxidative phosphorylation, so that normal proliferative mammal cells channel their final glycolytic metabolites into the mitochondria to meet their bioenergy demand [20]. Under constant glucose supply, proliferating cancer cells have been proposed to depend on aerobic glycolysis in order to provide glycolytic intermediates for excessive macromolecular biosynthesis and to rapidly produce ATP [21,22]. It should be noted that glucose level in human cancer tissue is much less than its physiologic concentration [19,23].

The following metabolic pathways are illustrated in Figure 1. There are three critical and irreversible catalytic steps in aerobic glycolysis [24,25]. The hexokinase (HK), the first rate-limiting step in glycolysis, regulates the intracellular glucose transport through glucose phosphorylation. HK converts glucose into glucose-6-phosphate (G6P), which proceeds along the glycolytic pathway or flux into the PPP depending on the cells metabolic demand. In the glycolytic stream, G6P is converted to fructose-6-phosphate (F6P), which is then catalyzed by the key rate-liming step phosphofructokinase-1 (PFK-1) into fructose-1,6-bisphosphate (F1,6BP). PFK-1, a gatekeeper of glycolysis, has three isoforms: PFKL (liver), PFKM (muscle), and PFKP (platelet) [26]. Cancer cells mainly utilize the PFKP for glycolysis and Snail suppresses PKFP to redirect glucose flux into PPP, resulting in NADPH production via oxidative branch [16]. While knockdown of Snail increases amino acid synthesis and lactate production, Snail-loss cancer cells are susceptible to cell death by oxidative stress or glucose starvation due to decreased NADPH level [16]. An additional critical enzyme activated in late glycolysis is pyruvate kinase M2, an isoform of pyruvate kinase that shifts the glucose flux to lactate formation in cancer cells contrary to physiologic aerobic mitochondrial respiration [27].

Gluconeogenesis shares reversible enzymes with glycolysis with two exemptions. FBP1 and glucose-6-phosphatase in gluconeogenesis are inverse reaction steps of PFK-1 and HK in glycolysis, respectively. FBP1 is the rate-limiting step in gluconeogenesis that either releases glucose into the blood stream or shifts the metabolites into the PPP [28,29]. In a physiologic condition, gluconeogenesis mainly occurs in liver and kidney under hypoglycemic condition, while FBP-1 is downregulated in specific types of human cancer [30,31]. Although FBP-1 is a Snail target gene, metabolic outcomes by loss of FBP1 in human cancer, whether increased mitochondrial flux or lactate production, are not yet well-understood, particularly under starved environment.

In general, counter-opposing PFK-1 and FBP1 are allosterically regulated by metabolite fructose 2,6-bisphosphate (F2,6BP), which is catalyzed by bifunctional PFK-2/FBP2 through glucose-associated hormones such as insulin, epinephrine, and glucagon. This indicates that F2,6BP contributes to glucose flux regulation [32,33]. F2,6BP increases the affinity of PFK-1 for F6P, enhancing glycolysis and inhibiting gluconeogenesis [34]. p53 downstream gene TP53-induced glycolysis and apoptosis regulator (TIGAR) has features similar to those of the bisphosphatase domain in FBP2 and degrades F2,6BP [35,36]. During cellular stress and early tumor suppressor gene p53 activation, TIGAR inhibits glycolysis, turning the carbon flux toward the PPP [35]. Therefore, activation of TIGAR and loss of PFK-1 play similar roles in terms of PPP flux.

## 4. Pentose Phosphate Pathway and NADPH Generation

The PPP is a main source for NADPH, which provides the major reducing equivalents for FAS and oxidative stress. Alternative NADPH, especially in glucose-deprived situations, is produced by FAO in the mitochondria [37]. PPP activation is essential in both the anabolic and catabolic pathways of cancer. It provides phosphopentoses and ribonucleotides for biosynthesis, as well as supporting FAS [38]. Oxidative phosphorylation highly relies on redox homeostasis as reactive oxygen species (ROS) are chiefly produced during mitochondrial activity [39,40]. The PPP is composed of two continuous branches, the oxidative branch, responsible for NADPH generation, and the non-oxidative branch, which provides reversible flows of glycolytic intermediates [5]. NADPH produced by oxidative PPP is also required for one-carbon folate metabolism [41]. The PPP is tightly regulated along with the glycolytic pathway and the carbon flux can either flow into the mitochondria or into gluconeogenesis and back to the PPP. This allosteric regulation gives flexibility to cancer cells in meeting the demands of metabolic reprogramming. Being a gatekeeper of glycolysis, PFPK plays a key role in regulating flux control between glycolysis and PPP. Interestingly, knockdown of PFKP in breast cancer cells induced dormancy-like cell cycle arrest with gain of resistance against oxidative stress, such as glucose starvation or paclitaxel treatment [16], indicating the importance of flux control between glycolysis and PPP. The glucose transport is shifted into the oxidative branch of PPP through G6P based on levels of ribulose-5-phosphate (Ribu5P), NADPH, and ATP as determined by metabolic demand [5,16,42]. The first product of the oxidative branch is Ribu5P, which can be isomerized and used as a nucleotide precursor or passed on to the non-oxidative branch. The non-oxidative branch of PPP is tightly interconnected with glycolysis and shares intermediate metabolites F6P and glyceraldehyde-3-phosphate (G3P or GADP). This reversible allosteric link is catalyzed by transketolase and transaldolase [43]. This flux can be recycled into the PPP though gluconeogenesis controlled by FBP1 to maintain the high NADPH levels required in physiologic tissue, such as liver, fat, and neural tissue [44]. Although the regulation of recycled flux in cancer is not understood in detail, metabolites may proceed into either upstream gluconeogenesis or the downstream TCA cycle based on oncogenic regulation of metabolic demand by bi-directional enzymes, such as aldolase and G3P dehydrogenase.

## 5. Fatty Acid Oxidation and ATP Formation

As glucose-deficient conditions impair the glycolytic pathway, cancer cells have to reduce their proliferation rate while increasing ATP and NADPH generation to promote survival. Due to lack of glucose intake, cancer cells acquire metabolites for oxidation from fatty acids, which are consumed in the citric acid cycle and oxidative phosphorylation to produce ATP. FAO, also known as beta-oxidation, is a catabolic countenance of the fatty acid metabolism and is linked with FAS in an exclusive manner [45]. FAS intermediate malonyl-CoA inhibits the main transport system, carnitine palmitoyltransferase 1 (CPT1), which transports exogenous or glucose-derived fatty acetyl-CoA into the mitochondria for beta-oxidation and later into oxidative phosphorylation [46]. Malonyl-CoA is produced via ACC, a gatekeeper for the reciprocal regulation between mutually exclusive FAO and FAS [47,48]. FAS requires substantial NADPH consumption for anabolic construction, while FAO principally produces ATP and NADPH. Therefore, in a glucose-starved condition, malonyl-CoA is a critical determinant of mitochondrial catabolic metabolism. During FAO, malonyl-CoA is controlled by ACC2 at the outer mitochondrial membrane, while ACC1 is located in the cytoplasm and mediates FAS [49]. In clinical samples, ACC2 was suppressed in various cancers, while ACC1 level was similar in neoplastic and normal tissue, indicating ACC2 activity and abundance comprise a core mitochondrial regulator in cancer [18]. Snail increased mitochondrial FAO and suppressed FAS by suppression of ACC2, reduced malonyl-CoA production, and subsequent increased CPT1 activity and TCA metabolites [18].

There has been growing evidence of the role of FAO during metastasis. FAO-associated genes are upregulated, inducing a metabolic shift during lymphatic metastasis [50]. Reduced intracellular ATP stimulates the liver kinase B1 (LKB1)-AMP-activated protein kinase (AMPK) pathway, which in turn inhibits ACC2. In addition, AMPK induces alternative NADPH for redox homeostasis in advanced glucose deficiency, wherein the PPP may be impaired [37]. In addition to glucose-depleted conditions, metastatic cells frequently encounter acidosis, which metabolically reprograms them toward FAO for adoption by mitochondrial hyperacetylation and ACC2 suppression [19,51,52]. These findings reinforce the significance of catabolic metabolism for tumor survival in stressful environments.

## 6. Paradox of Snail-Mediated Catabolic Metabolism

Snail is a main transcription factor of EMT, directly repressing E-cadherin transcription [7]. Snail transcript and protein abundance are regulated by oncogenic pathways such as Wnt, p53, transforming growth factor-β (TGF-β) superfamily, and receptor tyrosine kinase signaling [53,54,55,56,57,58]. In glucose-deficient conditions, which represent the metabolic status of the metastatic procedure, Snail specifically activates catabolic reprogramming. We have reported that Snail directly suppresses PFKP and ACC2 transcription, thereby switching the glycolytic flux into the PPP to enhance NADPH generation and stimulate oxidative phosphorylation via FAO, respectively [16,18]. ACC2 levels varied among breast cancer molecular subtypes in clinical samples, in an inverse relation to Snail with a reliance on p53 status. This implies that different clinical prognoses in molecular subtypes can be explained by Snail activity and p53 status, which we further investigated as described later in this article. In addition, other studies have shown that Snail inhibits FBP1 by DNA promoter methylation, a rate limiting enzyme of gluconeogenesis which has been proposed to enhance glycolysis [17,42,59,60]. As the gluconeogenesis by FBP1 is highly endergonic, the metabolic advantage by loss of FBP1 in cancer cells is quite clear. However, Snail paradoxically suppresses the rate limiting steps in glycolysis and gluconeogenesis via suppression of PFKP and FBP-1, respectively [16,42].

## 7. Breast Cancer Subtypes Use Different Metabolic Circuit to Gain Catabolic Advantages

Breast cancer is a heterogeneous spectrum of mammary gland-derived malignant tumors that can be subtyped by molecular profile in terms of standard-estrogen receptor (ER), progesterone receptor (PR), and human epidermal growth factor receptor 2 (HER2). Combinations of the three molecular profiles yield subtypes with clinically divergent implications: luminal A (ER+, PR+/−, HER2−, low Ki-67), luminal B (ER+, PR+/−, HER2+/−, high Ki-67), HER2 overexpression (ER−, PR−, HER2+), and basal-like (ER-, PR-, HER2-) [61,62]. The basal-like subtype constitutes the majority of the triple negative cases and generally has a worse prognosis than the luminal A or B subtype [61,63,64,65]. The basal-like subtype shows a higher p53-loss rate along with increased Snail abundance than do the luminal subtypes (mainly p53 wild type) [18,61,66]. As the triple negative breast cancers (TNBC) have an aggressive clinical course and lack treatment options, there have been various attempts to unravel their pathogenesis. EMT markers, including Snail, are highly evident in TNBC [18,67,68,69,70,71]. It should not be assumed that differentiated luminal subtypes lack EMT gene expression, but rather the EMT markers are selectively expressed and dedifferentiated [72]. The luminal B subtype has a high proliferative profile that typically opposes EMT activity [73]. Nevertheless, luminal subtypes have to enhance EMT transcription factors to overcome environmental stress and promote metastasis just as does TNBC.

The paradoxical suppression of PFKP and FBP1 by Snail suggested molecular subsets of different catabolic metabolism in breast cancer, leading us to further investigate metabolic-dependency based on breast cancer subtypes. We analyzed data from clinical samples from TCGA (The Cancer Genome Atlas) to determine transcript abundance of PFKP and FBP1 in breast cancer (BRCA) according to intrinsic subtype and mutational status of BRCA1/2, PIK3CA, and p53. Interestingly, clustering analysis with PFKP and FBP1 revealed distinct breast cancer subtypes, and transcript abundance of the two genes were inversely correlated (Figure 2A). The PFKP is mainly suppressed in luminal types having PIK3CA mutation while the FBP1 is suppressed in most basal-like types of breast cancer having p53 mutation, indicating a different catabolic circuit according to subtype and mutational status of PIK3CA and p53. While breast cancers having the BRCA mutation tend to favor the glycolytic pathway, the trend is not conclusive due to the limited number of BRCA mutation samples (Figure 2B,C). However, PIK3CA mutation in breast cancer was accompanied by lowered PFKP and higher FBP1, while p53 mutation revealed higher PFKP and lowed FBP1 (Figure 2D,E). These suggest that luminal types of p53 wt and PIK3CA mut breast cancer actively utilize PPP via suppression of PFKP and activation of FBP1 (Figure 2F). It should be noted that activation of gluconeogenesis may efficiently increase PPP flux [74]. In contrast, TNBC having p53 mutation and low frequency of PIK3CA mutation may have high glycolytic activity. These results indicate that EMT-mediated catabolic metabolism depends on breast cancer subtypes and oncogenic mutational status.

As the ACC2 comprises a Snail target resulting in increased FAO [18], we speculated that an FBP1-loss subtype, such as TNBC, is accompanied by suppression of ACC2 in breast cancer to support efficient mitochondrial catabolic metabolism. Interestingly, FBP1 and ACC2 transcript abundance were correlated, and ACC2 is suppressed in FBP1-loss basal-like subtype of breast cancer having p53 mutation (Figure 3A). Previously, we reported that ACC2 abundance is inversely correlated in higher Snail expression breast cancer samples having p53 mutation [18]. However, ACC2 abundance was correlated to neither PIK3CA nor BRCA mutation (Figure 3B). These results suggest that TNBC showing high glycolytic activity actively utilizes mitochondrial FAO. As noted, increased expression of TIGAR activates PPP flux. Analyzing TIGAR expression according to breast cancer subtypes, we found that TIGAR is increased in the FBP1-loss basal-like subtype of breast cancer (Figure 4). These results suggest that PPP is still important in TNBC regardless of glycolysis and mitochondrial-dependent catabolic metabolism [74].

## 8. Conclusions

In conclusion, metabolic reprogramming toward catabolism is essential for cancer cell survival and can be separately achieved in respective intrinsic subtypes of breast cancer via different opposing oncogenic regulations of Snail (Figure 5). The catabolic metabolism of luminal types of breast cancer largely depends on resulting PPP flux, while the basal-like subtypes actively utilize mitochondrial FAO. Our results provide diverse circuits of catabolic metabolism in human cancer and a novel therapeutic approach involving metabolic targeting according to cancer subtype.

## Figures and Tables

**Figure 1 cells-09-02064-f001:**
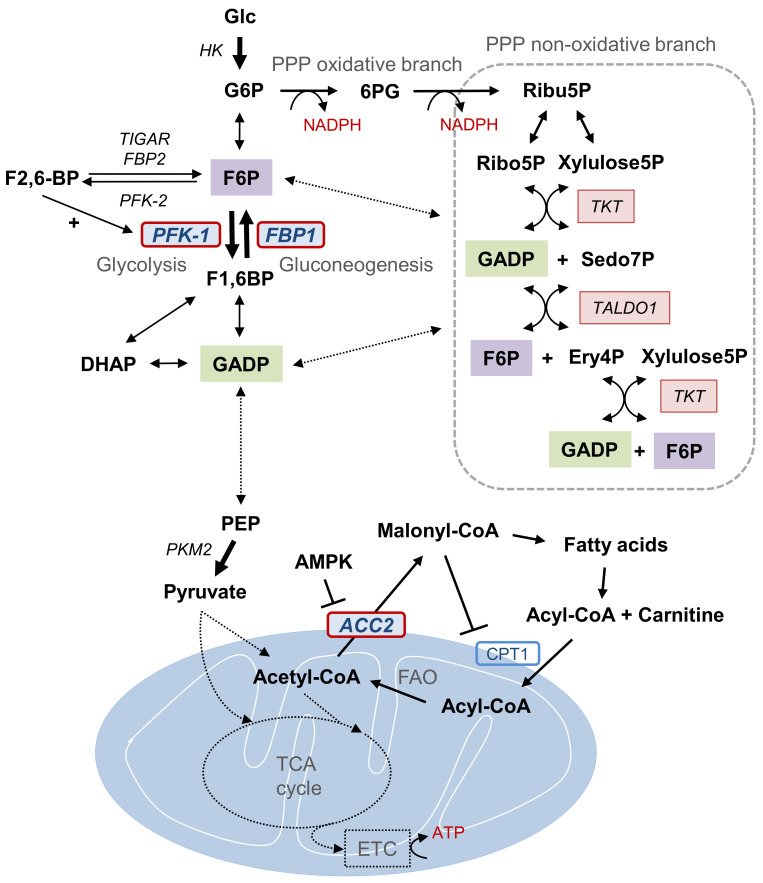
Metabolic pathways and Snail target enzymes of catabolic metabolism. Representative metabolic intermediates and enzymes of PPP, FAO, and the glycolytic pathway that constitute cancer cell catabolic metabolism in glucose-derived conditions are illustrated. Key metabolic enzymes (PFK-1, FBP1, and ACC2) targeted by EMT transcription factor Snail during catabolism are highlighted. Metabolic interaction between PPP, FAO, and the glycolytic pathway via key enzyme regulation induces the overall carbon flux toward catabolism and promotes cell survival via ATP and NADPH production. Abbreviations: PPP, pentose phosphate pathway; Glc, glucose; G6P, glucose 6-phosphate; 6PG, 6-phosphogluconate; Ribu5P, ribulose-5-phosphate; NADPH, reduced nicotinamide adenine dinucleotide phosphate; Ribo5P, ribose-5-phosphate; Xylulose5P, Xylulose 5-phosphate; GADP, glyceraldehyde 3-phosphate; TKT, transketolase; TALDO1, transaldolase 1; Sedo7P, sedoheptulose-7P; Ery4P, erythrose 4-phosphate; F6P, fructose 6-phosphate; F2,6-BP, fructose 2,6-bisphosphate; TIGAR, TP53-induced glycolysis and apoptosis regulator; PFK, phosphofructokinase; FBP, fructose 1,6-bisphosphatase; F1,6BP, fructose 1,6-bisphosphate; DHAP, dihydroxyacetone phosphate; PEP, phosphoenolpyruvate; PKM2, pyruvate kinase M2; AMK, AMP-activated protein kinase, ACC2, acetyl-CoA carboxylase 2; CPT1, carnitine palmitoyltransferase I; FAO, fatty acid oxidation; TCA, tricarboxylic acid.

**Figure 2 cells-09-02064-f002:**
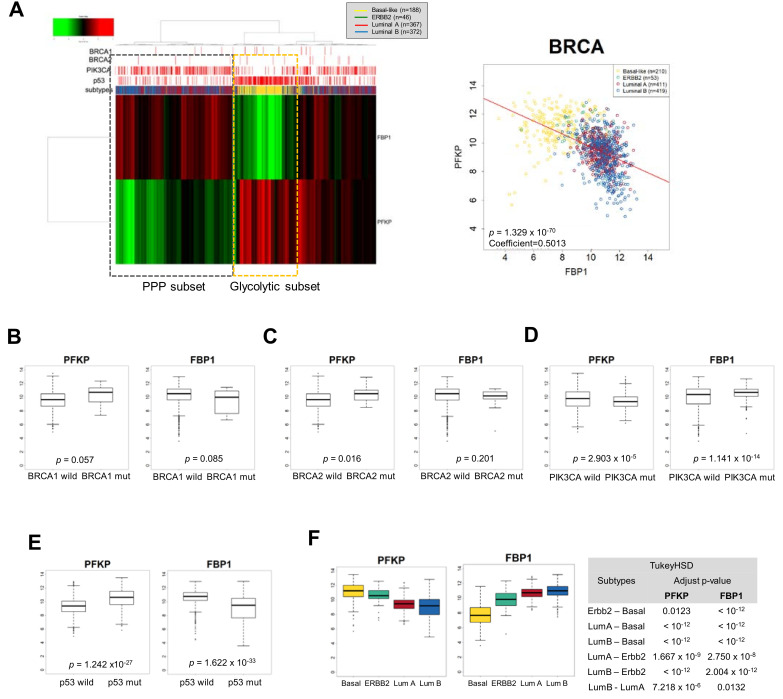
Mutual exclusive and differential expression of PFKP and FBP1 according to the breast cancer subtypes. (**A**) Unsupervised hierarchical cluster analysis of breast cancer patient samples in The Cancer Genome Atlas (TCGA) using PFKP and FBP1 to discriminate metabolic subsets with mutational status of BRCA1/2, PIK3CA, and p53 (left panel). Luminal subtypes show PFKP suppression with PIK3CA mutation (PPP dependent subset), while basal-like subtype shows FBP1 suppression (glycolytic subset) having p53 mutation. Pearson distance analysis with a negative distance correlation of PFKP and FBP1 with divergent clusters of breast cancer subtypes (right panel, coefficient = −0.501, *p* < 0.001). (**B**–**F**) Comparison of PFKP or FBP1 transcript abundances according to mutational status of BRCA1 (wt, *n* = 956; mut, *n* = 17) (**B**), BRCA2 (wt, *n* = 956; mut, *n* = 17) (**C**), PIK3CA (wt, *n* = 657; mut, *n* = 316) (**D**), and p53 (*n* = 675; mut, *n* = 298) (**E**) and breast cancer subtypes (**F**). Statistical significance was determined by Tukey’s HSD (honestly significant difference) test.

**Figure 3 cells-09-02064-f003:**
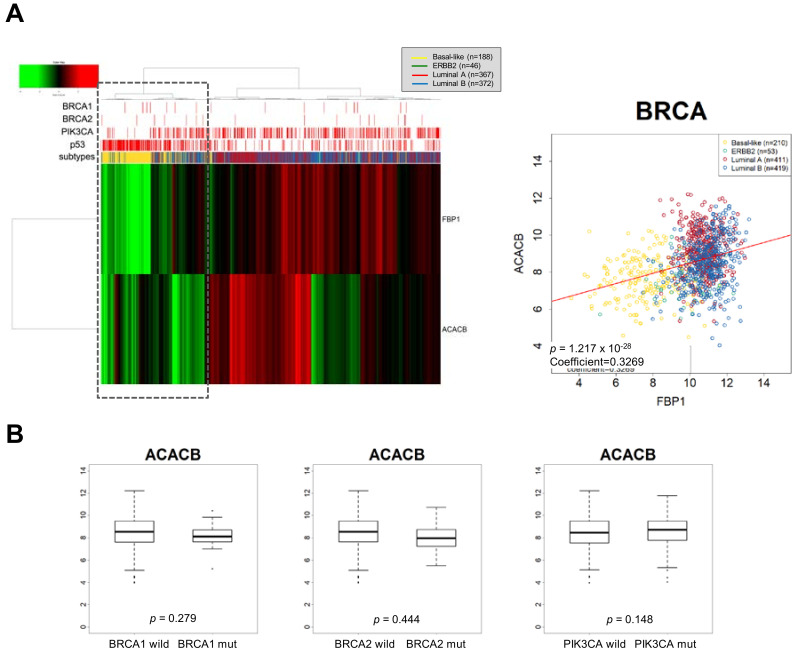
FBP-loss accompanies activation of FAO by down-regulation of ACC2 in basal-like subtype. (**A**) Unsupervised hierarchical cluster analysis of breast cancer patient samples in The Cancer Genome Atlas (TCGA) using FBP1 and ACC2 (ACACB) to discriminate between metabolic subsets of tumors with breast cancer subtypes (left panel). FBP1 and ACC2 were simultaneously suppressed in basal-like subtype having p53 mutation (dot lined box). Pearson distance analysis with a positive distance correlation of FBP1 and ACC2 (right panel, coefficient = −0.327, *p* < 0.001). (**B**) ACC2 transcript abundance according to the BRCA1/2 or PIK3CA mutation in breast cancer samples.

**Figure 4 cells-09-02064-f004:**
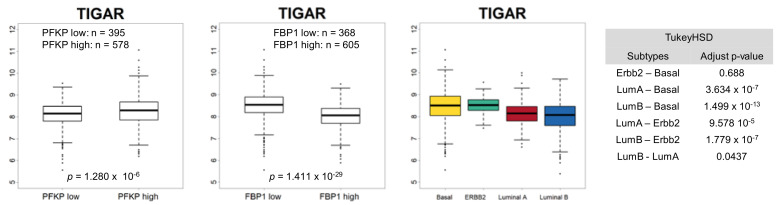
TIGAR is increased in FBP1-loss basal-like subtype of breast cancer. Comparison of TIGAR transcript levels according to PFKP (left panel), FBP1 (middle panel), and breast cancer subtypes (right panels). TIGAR expression was significantly higher in PFKP high, FBP1 low subsets, and basal-like subtype. Statistical significance was determined by Tukey’s HSD (honestly significant difference) test.

**Figure 5 cells-09-02064-f005:**
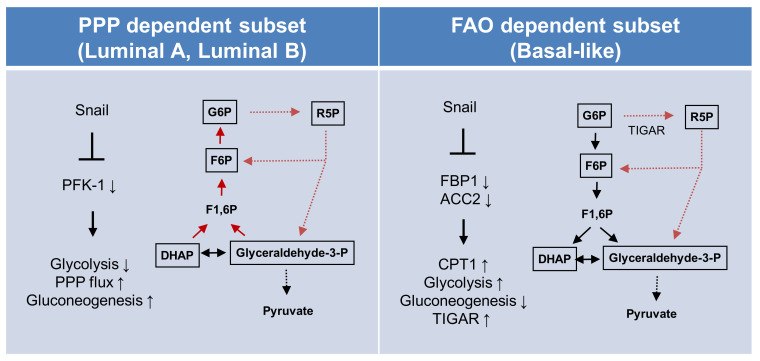
Proposed model of EMT-mediated catabolic metabolism in breast cancer subtypes. Proposed divergent metabolic models of PPP-dependent luminal subtype (left) and a FAO-dependent basal-like subtype (right) of breast cancer indicate Snail involvement in catabolic metabolism and cancer cell survival. Abbreviations: PPP, pentose phosphate pathway; FAO, fatty acid oxidation; ACC2, acetyl-CoA carboxylase 2; CPT1, carnitine palmitoyltransferase I; G6P, glucose 6-phosphate; R5P, ribose-5-phosphate; F6P, fructose 6-phosphate; TIGAR, TP53-induced glycolysis and apoptosis regulator; PFK-1 phosphofructokinase-1; FBP1, fructose 1,6-bisphosphatase 1; F1,6BP, fructose 1,6-bisphosphate; DHAP, dihydroxyacetone phosphate.

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
