# Peer review of "Breast Cancer Subtypes Underlying EMT-Mediated Catabolic Metabolism"

_cells, 2020, doi:10.3390/cells9092064_

Round 1
Reviewer 1 Report
In this article, the authors investigated the coordination of metabolic pathways expressed in breast cancer molecular subtypes underlying EMT-regulated catabolic metabolism. The authors focused on Snail as the inducer of EMT and regulator of catabolic metabolism via PFK1, FBP1, and ACC2.
The authors use several data inputs related to PFKP and FBP1 (Fig.2), FBP1 and ACC2 9 (Fig.3) and Tigar (Fig.4), but do not provide any data input related to Snail. Connection to Snail in presented dataset is performed by two approaches – 1. Literature citation of PFKP/FBP1 regulation by Snail 2. Analysis of current dataset by different subtypes of breast cancer and Snail expression in different subtypes (literature citation).
Section “2.6. Breast Cancer Subtypes Use Different Metabolic Circuit to Gain Catabolic Advantages” can benefit from additional clarification and expansion on “Snail expression in different subtypes of breast cancer”.
Statement earlier in the manuscript to establish logical chain that different subtypes of cancer = different Snail expression = our dataset with different subtypes of cancer would make it easier for readers to establish the connection to presented data.
Author Response
We appreciate the helpful comments on our manuscript. Because we have previously reported Snail abundance according to the breast cancer subtypes and p53 mutational status (please see Supplementary Fig. 2 in ref 18), we added a brief description of Snail abundance by breast cancer subtypes and p53 status (highlights, line 56-57; line 186-189; line 210-212).
Reviewer 2 Report
Cho ES et.al, demonstrates that EMT mediated catabolic metabolites dictates the breast cancer subtypes. These conclusions are based on mRNA seq data from breast cancer patients. Although this topic is interesting, but the experimental evidence is still lacking such as energy consumption versus production rate in breast cancer subtypes. SNAI1 expression in luminal/basal/HER2/claudin low how does it changes?
Major concerns:
- The rationale for SNAI1 and catabolites are missing, and experimental evidence is not shown.
- Does this catabolite derived subtypes have any association with the BRCA/P53/PIC3CA mutations?
- It is always interesting to know the metabolic and catabolic difference in the metabolic versus catabolic differences in the breast cancer subtypes.
Author Response
We appreciate the constructive comments on our manuscript. Because the role of Snail on catabolites and its functional relevance have been extensively studied previously (ref 16, 18, and 42), we have added detailed comments on catabolites in the revised manuscript with reference citation (highlights, line 11-15; line 135-139; line 167-170; line 203-204).
We also further analyzed the breast cancer subtypes in association with BRCA/p53/PIK3CA mutational status. Interestingly, p53 and PIK3CA mutation accompanied different catabolic subtypes of TNBC and Luminal types, respectively. We described these findings in the revised manuscript (line 216 - 230; line 243 - 250). We appreciate the reviewer’s insightful comment.
Round 2
Reviewer 2 Report
The authors have addressed my concerns and do not have any further comments.